

# Dynamic data transmission technology for expendable current profiler based on low-voltage differential signaling

Shuhan Li[1,2], Qisheng Zhang[1,2], Xiao Zhao[1,2], Shenghui Liu[1,2], Zhenzhong Yuan[1,2], Xinyue Zhang[1,2]

[1]Key Laboratory of Geo-detection (China University of Geosciences, Beijing), Ministry of Education
[2]School of Geophysics and Information Technology, China University of Geosciences, Beijing 100083, P.R. China

*Correspondence to:* Qisheng Zhang (zqs@cugb.edu.cn)

**Abstract.** A dynamic data transmission technology for expendable current profilers is proposed in this paper. Two parallel varnished wires are employed as the data transmission medium. By testing the transmission properties of the varnished wires, a baseband transmission system is studied and designed. Modified low-voltage differential signaling is adopted as the

physical layer for data transmission. The data transmission protocol is modified and optimized in accordance with the RS-232 protocol, and the Manchester code is superimposed. According to the results of indoor and marine tests, the data transmission distance of the designed system, which employs a 0.1mm diameter varnished wire, extends to 2 km with high efficiency and accuracy for data transmission, exhibiting excellent performance. Moreover, this data transmission technology could be used for other expendable marine-environment parametric measuring instruments such as an

expendable bathythermograph and an expendable conductivity temperature depth profiler.

## 1 Introduction

An expendable current profiler (XCP) is a type of marine-environment measuring device that can rapidly obtain information about current profiles (Liu and He, 2010). An XCP probe can be launched from a carrying platform, such as a ship, submarine, or aircraft. As the probe sinks, the current and temperature profiles are rapidly measured. In addition, the water

depth can be calculated from the speed at which the probe sinks (Liu et al., 2007; Chen et al., 2011). The measured data are transmitted back to the carrying platform through wireless communication (Abdullah and Bong, 2014; Wang and Gulliver, 2015) and are then processed to determine, in real time, the variation in the dynamic marine-environment parameters with the water depth. Thanks to these capabilities, XCP is used as an advanced and efficient measuring method for application to scientific marine investigations, environmental investigations for fisheries, and marine environmental observations for the

military (Zhang et al., 2013; Liu and Chen, 2011; Liu et al., 2010).

In 1971, Sanford proposed a fundamental calculation formula for measuring the electromagnetic fields induced by the movement of water in the ocean. He and his colleagues successfully developed the first expendable temperature and velocity profiler in 1978 (Sanford, 1971; Sanford et al., 1978). Sippican, Inc., took the design of this expendable temperature and velocity profiler, renamed it an "expendable current profiler", and put it into mass production. It was employed in the



drafting of the Antarctic Circumpolar Current profile (Sanford, 1982; Szuts, 2012) and in the detection of the rapid overflow phenomenon in the Denmark Strait (Girton et al., 2001).

A problem with expendable marine-environment measuring profilers is the long distances over which dynamic data transmission must be carried out. The specialty of dynamic transmission is the progressive lengthening of the transmission

line as the probe descends into the ocean in a spin-stabilized manner. This requires the transmission system to have powerful data transmission driving and interference resistance capabilities. As the probe signals are collected continuously while the probe sinks rapidly, a wide bandwidth is required by the transmission system to handle the large quantities of data that are being transmitted. Dynamic data transmission from the probe to the float is frequently a key factor restricting the progress of expendable measuring profilers (Qian et al., 2016). Experiments have demonstrated that seawater has a significant impact on

electrical signal transmission over varnished wires. Instruments based on analog signal transmission usually experience a decrease in their precision as a result of noise. As part of our development of XCP, digital signal transmission was employed as a means of solving the data transmission problem associated with varnished wires. This approach eventually produced significant progress. At present, the transmission speed of the XCP is limited to 4800 bps, while the maximum transmission distance is 2 km. In this paper, the issues affecting digital signal dynamic transmission between an XCP probe and float are

discussed in detail.

## 2 Performance testing of varnished wire

The XCP developed in this study consists of three parts: a probe, float (which acts as a transfer unit), and wireless receiver (on board the carrying platform), as shown in Fig. 1. The data transmission medium between the probe and the float consists of two varnished wires, arranged in parallel, with an outer diameter of 0.1 mm and a weight of 15 g per 100 m. The

resistance of the varnished wire is $1.91\ \Omega\ m^{-1}$. The two varnished wires are arranged in parallel and are tightly bound to each other. These two bound wires are encased together in a highly insulating varnish. As the surfaces of each wire have been varnished, there is no need to worry about interference between the two bound wires. This varnished wire is well suited to use in expendable measuring profilers owing to its low weight and cost. To further examine the data transmission performance of the varnished wire, an SA1030 frequency-scanning meter was used to test the amplitude–frequency

characteristics of 2 km of varnished wire in air. The results are summarized in Fig. 2. The SA1030 utilized in our study has an output impedance of 50 Ω and a high input impedance. The output signal used for the test was a 1V peak-to-peak sine wave.

As shown in Fig. 2, the amplitude of the signal passing through the varnished wire rapidly decreases as the signal frequency increases, indicating that the varnished wire acts as a typical low-pass channel. The amount of signal information to be

transmitted is much larger than that of the channel capacity in the transmission bands. Hence, to ensure signal integrity, signal transmission must be conducted in the cut-off frequency band, which means that the frequency response of the


transmission channel must be improved. A corresponding compensation method can be used to expand the cut-off frequency band of the varnished wire into a high-frequency band to satisfy the dynamic data transmission requirements.

**3 Transmission system physical layer**

After varnished wire had been selected as the data transmission medium, the transmission system physical layer was then

determined. Low-voltage differential signaling (LVDS) was ultimately selected. However, before this selection was made, a controller area network (CAN bus) was tested for the physical layer. The indoor test results showed that the CAN bus—a bus technology with an apparent data transmission distance of 10 km (Tong et al., 2007; Yu, 2001; Sun et al., 2008)—could not transmit data in a static situation even over a 1m varnished wire. This indicates that data transmission through this type of varnished wire is difficult. Therefore, a LVDS sender configured on a programmable chip was selected for the physical

layer at the sending end, as shown in Fig. 3.

LVDS is a type of ground-isolated differential signaling with a strong anti-interference ability. It has a relatively strong current drive capability, which makes it appropriate for high-speed, long-distance, and low-power dissipation transmission. In a digital system, a two-element digital signal is transmitted. Because a binary pulse signal without modulation typically starts from a direct current or low frequency, it is called the baseband signal (Fan, 2001). In a baseband signal transmission

system, channel distortion is relatively serious. It primarily includes amplitude distortion and group delay distortion. For non-distortion transmission, a channel equalizer is typically attached to the receiving end of the float. The channel equalizer is an isolation transformer that was made specifically for communication. We hand-wound it ourselves. It has two main functions: turning the transmission channel into a bandwidth-limited band-pass channel, and equalizing the amplitude to help straighten the frequency–amplitude curves in the channel band.

In an actual operational setting, the varnished wire is rapidly and dynamically line cast. In the early stages of this line casting, most of the varnished wire is entwined, such that induction arises. When it is placed in high-permittivity seawater (Zhang et al., 2013; Dong et al., 2009), it exhibits an increasingly strong capacitance. For the reasons mentioned above, the LVDS signal had to be overlaid with a large amount of noise. In addition, an obvious attenuation of the signal when transmitted over the varnished wire was observed. Therefore, a high-gain transformer was employed for the receiving end of the float.

The transformer provides noise isolation and signal amplification. Then, the combination of the high-pass and low-pass forms a band-pass to eliminate interference within the transmission channel. The cut-off frequency of the band-pass was found to be 0.2–8 kHz. After a hysteresis comparison and a scaling circuit, the original data to be transmitted were restored and extracted, as shown in Fig. 4.



**4 Transmission protocol and code pattern**

As mentioned above, the data transmission protocol used in the proposed design was developed and optimized in accordance with the RS-232 protocol (Vijaya et al., 2011). A data frame consists of a start bit, eight data bits, and a stop bit. During the data sending process, one data frame is sent at a time. This requires the sending and receiving baud rates to be consistent.

The receiver detected the signal at a speed that was 16 times greater than that of the data transmission baud rate. When an effective start bit was obtained, the eight data bits and stop bit were oversampled at a speed that was 16 times greater than the baud rate. The seventh, eighth, and ninth sampling values were recorded. By using the majority-vote method, these sampling values were sent to a shift register when the values of two or more samples were the same. When the register finished reading the eight data bits, the last digit—the stop bit—was read. When the stop bit was effective, this frame of data

was regarded as being effective, and the interrupt flag bit in the program was placed in the effective status. Otherwise, this frame of received data was abandoned, and the receiving circuit was reset for the effective start bit. Therefore, by calculating the number of lost bytes, we can derive the bit error rate (BER) for the data transmission.

By adopting the majority-vote method to improve the RS-232 protocol, and by leveraging the remarkable anti-jamming capability of serial communication, the anti-noise performance of the data transmission system was greatly improved.

Moreover, the production cost and complexity of the hardware decreased.

Given the lack of consideration afforded the frequency range for data transmission in the improved RS-232 protocol, a signal cannot be directly transmitted over certain band-pass channels. Therefore, the Manchester code was merged into the established process of the transmission system, as shown in Fig. 5. Manchester coding, also known as phase coding, is a synchronous clock encoding technique used by the physical layer to encode the clock and data of a synchronous bit stream.

In Manchester coding, there is a transition in the middle of each bit period. Transition from a high level to a low level indicates binary "1" whereas transition from a low level to a high level indicates binary "0". Each element is transformed into two levels.

Based on the improved RS-232 protocol, the data bits were encoded based on Manchester coding to decrease the size of the low-frequency direct current part of the transmitted data. This approach was more beneficial to data transmission over

varnished wire. Redundancy was added to the Manchester coding to enable the data to be proofread and corrected while being decoded (Ogundile et al., 2015) at the receiving end and to conduct statistical calculations. Corrections were implemented by calculating the forward and backward code elements. The coded signal had an energy distribution mainly within a range of 0.5 to 7 kHz at the transmission speed of 4800 bps, which is the frequency band that can be used by the transmission system.




**5 Marine tests of transmission system**

To test the data transmission performance of the varnished wire in a real marine environment, we performed both a marine test of data transmission over varnished wire and a marine test of the entire XCP system with the varnished wire in the South China Sea.

5    In April 2009, marine tests were undertaken to investigate data transmission over the varnished wire by using a wire with a total length of 1988 m. During the test, the varnished wire was connected to the transmitter at one end and to the receiver at the other end. The varnished wire was gradually lowered into the seawater while data were being sent by the transmitter to the receiver. Both the transmitter and receiver were commercially available modules purchased through regular commercial channels. The test results show that, out of a total of 768,786 data sets there were 1553 errors, corresponding to a BER of 10    0.202 %, at a transmission speed of 4800 bps.

After finishing the marine test of the data transmission over the varnished wire, the varnished wire was applied to our developed XCP when we implemented our marine test in October 2009. During this test, the ocean current velocity and temperature were measured, and the results show that our developed XCP can meet the measurement requirements of the current profile parameters. The data obtained in the XCP marine test verified the effectiveness and practicality of the 15    dynamic data transmission technology described in this paper.

**6 Conclusions**

This study set out to analyze dynamic data transmission technology for XCP. The results of our research and testing allow us to draw the following conclusions: (1) The insulating property of the varnished wire has a critical effect on the transmission distance, with better insulation resulting in improved transmission quality. (2) In actual marine-environment tests, the 20    transmission system is capable of reliable transmission over distances of 1650 m (or even 2 km in optimum cases) at a transmission speed of 4800 bps. (3) The transmission technology described in this paper can be applied not only to XCP but also to other expendable instruments used to measure marine environmental parameters. For example, this technology has been adopted and used in both an expendable bathythermograph (XBT) and expendable conductivity temperature depth profiler (XCTD) at the Institute of Oceanographic Instrumentation, Shandong Academy of Sciences. Its use has helped to 25    overcome a data transmission problem that has long been an issue.

If transmission distances are to exceed 2 km while maintaining a transmission speed of 4800 bps, an in-depth study will be necessary to improve the hardware circuits and thus enhance the performance, while the data transmission protocol must be optimized through further analysis.

30



*Acknowledgments.* This work was supported by the Open Fund (No. GDL1611) of Key Laboratory of Geo-detection (China University of Geosciences, Beijing), Ministry of Education, the Natural Science Foundation of China (No. 41574131), the National "863" Program of China (No. 2012AA061102), and the Fundamental Research Funds for the Central Universities of China (No. 2652015213).

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

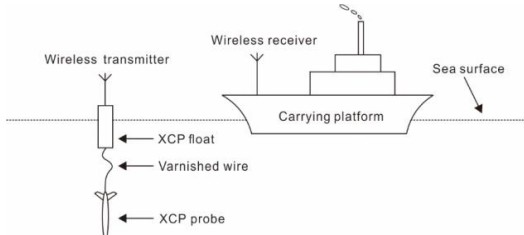

**Figure 1: Schematic diagram of expendable current profiler.**

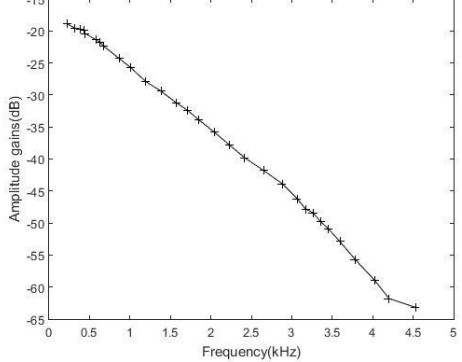

**Figure 2: Amplitude–frequency characteristics of varnished wires in air.**



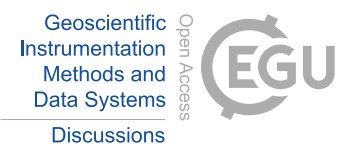

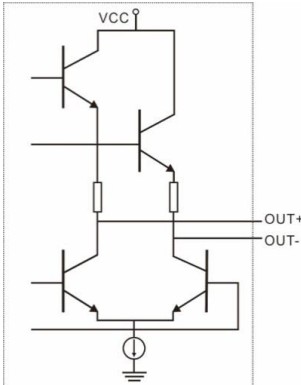

**Figure 3: Physical layer structure of sender.**

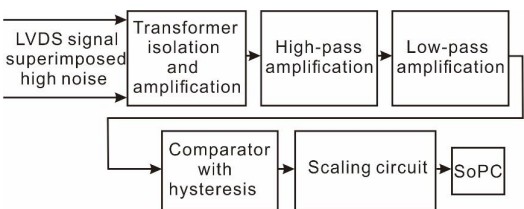

5    **Figure 4: Physical layer structure of floating receiving side.**

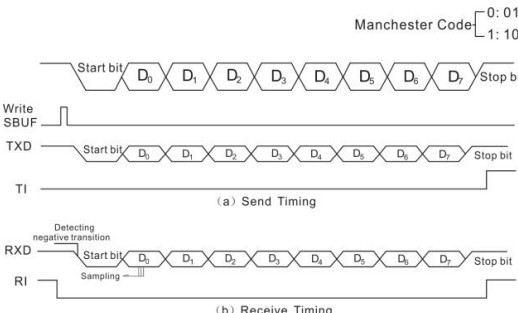

**Figure 5: Superposition of Manchester encoding data-transmission sequence diagram.**