# Peer review of "Dynamic data transmission technology for expendable current profiler based on low-voltage differential signaling"

_Geoscientific Instrumentation, Methods and Data Systems, 2017_

## Referee Comment (RC1) · C. Guangyuan (Referee) · 11 Apr 2017

C. Guangyuan (Referee)

freebird0430@163.com

This manuscript focuses on the research on an XCP data transmission technology which is original and meaningful in this field. It falls into the scope of GI and is scientifically sound to be published in its discussion forum. But there are still some minor revisions should be done. Please follow the comments below and revise your manuscript carefully.

(1) In this paper, the word "float" appears many times, maybe the word "buoy" is more appropriate than "float".

(2) In Fig. 4, the abbreviation "SoPC" should be defined or explained in the manuscript.

[Figure]

(3) In p. 4, line 5. In the proposed receiving technique, the majority vote method based three sampled signal level near the center of the bit. The authors should explain comparison with other method with additional parameters such as five or seven samples. It should be shown that the parameter of three samples is optimum.

(4) In the part of conclusion, the BER of data should be added.

Please also note the supplement to this comment:
http://www.geosci-instrum-method-data-syst-discuss.net/gi-2017-6/gi-2017-6-RC1-supplement.pdf

---

## Author Comment (AC1) · 12 Apr 2017

Dear Chen Guangyuan,

Thank you very much for your precious time to spend on our manuscript. Your comments are all valuable and very helpful for revising and improving our paper. Based on your comments, we have checked the manuscript carefully and made necessary revisions accordingly. We herewith provide our revisions as below:

(1) According to your suggestion, the word "float" has been replaced by "buoy", and relevant modifications have been made to Figure 1.

(2) Based on your suggestion, we have added the explanation to the abbreviation "SoPC" in the manuscript.

(3) We are aware that the value sampled close to the center of each data bit will be the most representative of the data bit's true value. The receiver detected the signal at a speed that was 16 times that of the data transmission baud rate, and the seventh, eighth, and ninth sampling values were recorded near the center of the bit. In our research, we have tested the use of 5 and 9 sampling points and found that the resulting measurement efficacies were the same as having only 3 sampling points. As the selection and judgment of the data bits were performed within the program, the time taken by the program for judgment increases with the number of selected sampling points, which decreases the efficiency of data transmission. Hence, we found the selection of 3 sampling points to be more appropriate for our purposes.

(4) According to your suggestion, the BER of data has been added in the part of conclusion.

---

## Referee Comment (RC2) · Anonymous Referee #2 · 13 Apr 2017

This manuscript is meaningful for GI field, but still have some grammar mistakes should be checked again.

---

## Author Comment (AC2) · 16 May 2017

Dear Referee #2,

Thank you very much for your precious time to spend on our manuscript. Your comments are all valuable and very helpful for revising and improving our paper. According to your comments, we have checked the manuscript carefully and made necessary revisions accordingly.